# PCOS and the Genome: Is the Genetic Puzzle Still Worth Solving?

**DOI:** 10.3390/biomedicines13081912

**Published:** 2025-08-05

**Authors:** Mario Palumbo, Luigi Della Corte, Dario Colacurci, Mario Ascione, Giuseppe D’Angelo, Giorgio Maria Baldini, Pierluigi Giampaolino, Giuseppe Bifulco

**Affiliations:** 1Department of Public Health, School of Medicine, University of Naples “Federico II”, 80131 Naples, Italy; dario.colacurci@unina.it (D.C.); mario.ascione@unina.it (M.A.); giuseppe.dangelo3@unina.it (G.D.); pierluigi.giampaolino@unina.it (P.G.); giuseppe.bifulco@unina.it (G.B.); 2Department of Neuroscience, Reproductive Sciences and Dentistry, School of Medicine, University of Naples “Federico II”, 80131 Naples, Italy; luigi.dellacorte@unina.it; 3Department of Biomedical Sciences and Human Oncology, University of Bari “Aldo Moro”, Piazza Giulio Cesare 11, 70124 Bari, Italy; giorgio.baldini@uniba.it

**Keywords:** polycystic ovary syndrome, genetics, epigenetics, gene–environment interaction, DNA methylation, non-coding RNAs, insulin resistance, ovarian dysfunction, endocrine disorders, reproductive health

## Abstract

**Background:** Polycystic ovary syndrome (PCOS) is a complex and multifactorial disorder affecting reproductive, endocrine, and metabolic functions in women of reproductive age. While environmental and lifestyle factors play a role, increasing evidence highlights the contribution of genetic and epigenetic mechanisms to its pathogenesis. **Objective:** This narrative review aims to provide an updated overview of the current evidence regarding the role of genetic variants, gene expression patterns, and epigenetic modifications in the etiopathogenesis of PCOS, with a focus on their impact on ovarian function, fertility, and systemic alterations. **Methods:** A comprehensive search was conducted across MEDLINE, EMBASE, PubMed, Web of Science, and the Cochrane Library using MeSH terms including “PCOS”, “Genes involved in PCOS”, and “Etiopathogenesis of PCOS” from January 2015 to June 2025. The selection process followed the SANRA quality criteria for narrative reviews. Seventeen studies published in English were included, focusing on original data regarding gene expression, polymorphisms, and epigenetic changes associated with PCOS. **Results:** The studies analyzed revealed a wide array of molecular alterations in PCOS, including the dysregulation of *SIRT* and estrogen receptor genes, altered transcriptome profiles in cumulus cells, and the involvement of long non-coding RNAs and circular RNAs in granulosa cell function and endometrial receptivity. Epigenetic mechanisms such as the DNA methylation of *TGF-β1* and inflammation-related signaling pathways (e.g., *TLR4/NF-κB/NLRP3*) were also implicated. Some genetic variants—particularly in *DENND1A*, *THADA*, and *MTNR1B*—exhibit signs of positive evolutionary selection, suggesting possible ancestral adaptive roles. **Conclusions:** PCOS is increasingly recognized as a syndrome with a strong genetic and epigenetic background. The identification of specific molecular signatures holds promise for the development of personalized diagnostic markers and therapeutic targets. Future research should focus on large-scale genomic studies and functional validation to better understand gene–environment interactions and their influence on phenotypic variability in PCOS.

## 1. Introduction

Polycystic ovary syndrome (PCOS) is one of the most prevalent endocrine disorders affecting women of reproductive age, with an estimated global prevalence ranging from 3% to 15% [1]. PCOS is clinically characterized by a constellation of symptoms, including ovulatory dysfunction, hyperandrogenism, and insulin resistance (IR). These features impair fertility and also increase the risk of long-term metabolic and cardiovascular complications, including type 2 diabetes and atherosclerosis. Despite extensive clinical recognition, the exact etiology and pathophysiological mechanisms of PCOS remain incompletely understood [2].

The diagnosis of PCOS is primarily based on the Rotterdam criteria, established in 2003, which require the presence of at least two out of three of the following features: (1) clinical and/or biochemical signs of hyperandrogenism, (2) ovulatory dysfunction (oligo- or anovulation), and (3) polycystic ovarian morphology (defined as ≥12 follicles measuring 2–9 mm in diameter and/or ovarian volume >10 mL on ultrasound) [3].

This classification allows for recognition of four distinct PCOS phenotypes, reflecting the heterogeneity of the syndrome [4].

The recent literature has proposed an expanded subcategorization, particularly focusing on patients with obesity and metabolic syndrome. These patients often present with elevated body mass index (BMI), increased waist-to-hip ratio (WHR), and higher scores on the homeostasis model assessment of insulin resistance (HOMA-IR). Such features are strongly associated with worsened endocrine and metabolic profiles, reinforcing the hypothesis of a metabolic phenotype of PCOS [3,4].

While the etiopathogenesis of PCOS remains unclear, it is widely accepted that the syndrome arises from a complex interplay between genetic predisposition and environmental factors, including diet, lifestyle, and exposure to endocrine-disrupting chemicals. The present work focuses on the investigation of internal molecular mechanisms, particularly the role of genetic and epigenetic factors that may predispose individuals to the development of PCOS [4,5].

Although no single causative gene has been definitively identified, several neuroendocrine abnormalities have been consistently observed in PCOS patients. These include increased the pulsatility of gonadotropin-releasing hormone (GnRH) secretion, leading to elevated luteinizing hormone (LH) levels, higher anti-Müllerian hormone (AMH) production, and a concomitant reduction in follicle-stimulating hormone (FSH) concentrations [5]. These hormonal imbalances contribute to disrupted folliculogenesis, anovulation, and excess androgen production by ovarian theca cells [5].

Another key area of interest is the relationship between insulin resistance, obesity, and ovulatory dysfunction. In many cases, metabolic alterations associated with hyperinsulinemia are more influential in the pathophysiology of anovulation than androgen excess. Insulin acts synergistically with LH to stimulate androgen production and inhibits hepatic synthesis of sex hormone-binding globulin (SHBG), thereby increasing free androgen levels [6].

PCOS is thus recognized as a multisystem disorder with reproductive, dermatological, and metabolic manifestations.

Clinical features may include menstrual irregularities (oligomenorrhea, amenorrhea), infertility, acne, hirsutism, alopecia, acanthosis nigricans, and other signs consistent with insulin resistance or metabolic syndrome. These features often coexist, further complicating diagnosis and management [7].

The aim of this review is to provide an updated and comprehensive overview of the aetiopathogenic role of genetic and molecular factors in PCOS, with a focus on identifying shared molecular pathways and gene variants that may underlie the development and heterogeneity of this syndrome.

## 2. Materials and Methods

This narrative review was conducted in accordance with the quality standards defined by the SANRA [8] (Scale for the Assessment of Narrative Review Articles) guidelines for methodological rigor and transparency. A comprehensive literature search was performed using the following electronic databases: MEDLINE, EMBASE, Web of Science, PubMed, and the Cochrane Library. The review protocol was prospectively registered in the PROSPERO database (registration number: CRD420241110847) to ensure methodological transparency and adherence to review standards.

The search strategy employed a combination of Medical Subject Headings (MeSH) and free-text terms, including “PCOS”, “genes involved in PCOS”, “etiopathogenesis of PCOS”, and “PCOS and genetics” from January 2015 to June 2025. Boolean operators (“AND”, “OR”) were used to optimize the sensitivity and specificity of the search.

Two reviewers (*MP* and *LDC*) independently screened all titles and abstracts retrieved through the initial search. All types of study designs were considered eligible, including observational studies, case–control studies, cohort studies, and genetic association studies. Articles deemed potentially relevant were selected for full-text review. Inclusion criteria required that studies be published in English and report original data related to the genetic or molecular mechanisms underlying PCOS.

Disagreements during the selection process were resolved through discussion, and, when necessary, a third reviewer (*DC*) was consulted to achieve consensus. In addition, two reviewers (*MA* and *GD*) manually reviewed the reference lists of all included articles to identify any studies that may have been missed in the database search.

Articles were excluded if they met any of the following criteria: (1) focused exclusively on in vitro or animal model studies without translational relevance to human PCOS; (2) were conference proceedings, abstracts, or non-peer-reviewed publications; or (3) did not report specific findings related to genetic or molecular aspects of PCOS.

Relevant data from each included study were extracted and synthesized narratively, with particular attention to the genes investigated, their molecular functions, the pathophysiological context, and the clinical relevance of the findings (Figure 1).

The selection process prioritized original studies that provided functional insights into the genetic and epigenetic mechanisms underlying PCOS. Specifically, we focused on studies involving gene expression analyses, epigenetic modifications, and mechanistic studies exploring the biological impact of specific variants or regulatory pathways. In contrast, studies limited to purely descriptive or statistical associations—such as SNP association studies without functional validation—were excluded. This approach was adopted to enhance the translational relevance of the findings and to better capture the molecular underpinnings of PCOS pathophysiology.

## 3. Advances in Genomic Research on PCOS

### 3.1. Understanding PCOS Through Genomic Approaches

Figure 1 illustrates the process of study selection. A total of 60 articles were initially retrieved from database searches. After title and abstract screening, 49 articles remained. Following a full-text evaluation, 35 studies were assessed for eligibility, and ultimately 17 studies met the inclusion criteria and were analyzed in detail.

Of the 17 included studies, 11 provided significant insights into the genetic and epigenetic landscape of PCOS (a summary is provided in Table 1).

The selected studies explored a broad spectrum of genetic expression patterns, single nucleotide polymorphisms (SNPs), epigenetic modifications, and their functional implications on ovarian function, endometrial receptivity, and systemic metabolic dysregulation in PCOS (Table 1).

In order to ensure a high level of scientific relevance, we applied a stringent selection process that favored studies offering innovative insights into the pathogenesis of PCOS—specifically those addressing the genetic and epigenetic mechanisms implicated in affected women. As a result, many studies with limited novelty or lacking functional relevance were excluded, which justifies the relatively low number of articles ultimately included in this review.

In the description of the studies analyzed, we began our methodological approach with a work that exemplifies the integration of molecular data and clinical phenotypes, serving as a conceptual starting point for our review. This allowed us to progressively build a narrative centered on functionally relevant and emerging genetic and epigenetic mechanisms contributing to PCOS pathogenesis.

González-Fernández et al. [9] investigated the expression of the sirtuin (SIRT) gene family in various gynecological conditions, including PCOS. These genes encode NAD^+^-dependent deacetylases involved in chromatin remodeling, cellular stress resistance, inflammation regulation (via NF-κB inhibition), and mitochondrial metabolism.

They reported increased *SIRT2* gene expression across all diagnostic categories, including PCOS, suggesting a possible compensatory or dysregulated metabolic response. However, the specific implications of *SIRT* gene dysregulation in oocyte competence and reproductive outcomes in PCOS remain to be fully elucidated.

In addition, mutations in the estrogen receptor genes *ESR1* and *ESR2*—key regulators of folliculogenesis and ovulation—were evaluated by Mucee et al. [10] Their study identified SNPs rs1467954450 (*ESR1*) and rs140630557 (*ESR2*) that impair receptor binding to estradiol, potentially altering estrogen signaling in granulosa cells. These variants may serve as genetic biomarkers for PCOS susceptibility and warrant further exploration in population-wide cohorts.

Ali Akbari et al. [11] analyzed gene expression in cumulus cells (CCs), revealing significant changes in the expression of *CALM1*, *PSMD6*, and *AK124742*. The upregulation of *CALM1* and *AK124742* and the downregulation of *PSMD6* were observed in PCOS patients. These alterations suggest a disrupted transcriptomic environment surrounding the oocyte, potentially affecting oocyte maturation and embryo viability.

Moreover, Zhiheng Yu et al. [12] proposed an evolutionary-genetic framework for PCOS. Their data revealed positive selection signatures at several PCOS-related loci, including *DENND1A*, *AOPEP*, *THADA*, *DGKI*, and *UNC5C*. These findings suggest that historical adaptive advantages may have contributed to the maintenance of PCOS risk alleles in human populations despite their current pathological manifestations.

Emerging evidence highlights the role of non-coding RNAs in PCOS, Zhao et al. [13] conducted a transcriptomic analysis of circular RNAs (circRNAs) in a PCOS mouse model. Their data revealed 205 differentially expressed circRNAs, implicating *circRNA_38548*, *circRNA_001686*, *circRNA_38550*, and *circRNA_27938* in the disruption of endometrial receptivity through miRNA sponging and the modulation of key target genes (e.g., *FOXO1*, *HOXA10*, *LIFR*). These findings suggest that altered circRNA-miRNA-mRNA networks may contribute to infertility in PCOS.

Furthermore, Jiang et al. [14] used bioinformatics and machine learning to identify two hub genes—*CXCR2* and *LMNB1*—as senescence-associated markers with diagnostic relevance for PCOS. These genes and their regulatory networks were linked to immune modulation and granulosa cell aging. Their results point to cellular senescence as a potential pathogenic mechanism in PCOS-related ovarian dysfunction.

*FoxO1*, a forkhead transcription factor, was found to be overexpressed in granulosa cells of PCOS rats, as reported by Huang et al. [15] *FoxO1* knockdown alleviated ovarian inflammation and fibrosis, promoted granulosa cell proliferation, and suppressed activation of the *TLR4/NF-κB/NLRP3* inflammasome pathway. These results indicate a potential therapeutic target for reducing ovarian inflammation in PCOS.

Gao et al. [16] investigated *TGF-β1* as a susceptibility gene in familial PCOS with insulin resistance (IR). Whole-exome sequencing and bisulfite methylation assays revealed hypomethylation at CpG sites in the *TGF-β1* promoter, correlating with increased gene expression and IR severity. These findings underscore the role of epigenetic mechanisms in metabolic disturbances associated with PCOS.

Monshizadeh et al. [17] assessed the expression of *BAMBI* and *RBX1* in cumulus cells from MII and GV-stage oocytes. The expression of both genes was significantly downregulated in PCOS patients, implicating their potential involvement in oocyte developmental competence. Postolache et al. [18] examined polymorphisms in melatonin receptor genes *MTNR1A* and *MTNR1B* in familial PCOS. Six variants were significantly associated with PCOS susceptibility. Given melatonin’s regulatory role in circadian rhythms and ovarian steroidogenesis, these receptor variants may influence ovulatory patterns and reproductive timing.

Yang et al. [19] identified *SNHG5* as a long non-coding RNA upregulated in granulosa cells of PCOS patients. The overexpression of *SNHG5* in vitro impaired cell proliferation and promoted apoptosis by sponging *miR-92a-3p* and upregulating *CDKN1C.* These findings provide novel insights into GC dysfunction and highlight *SNHG5* as a potential therapeutic target.

PCOS is a multifactorial disorder resulting from a complex interplay between genetic, epigenetic, and environmental factors. Advances in high-throughput technologies and integrative omics approaches have unveiled a growing number of candidate genes and molecular pathways potentially implicated in its pathogenesis.

Moreover, epigenetic modifications—shaped by both intrinsic and extrinsic influences, appear to mediate long-term changes in gene expression relevant to reproductive and metabolic dysfunctions. Understanding these interconnections is crucial to identifying novel biomarkers and therapeutic targets.

Together, the studies summarized in Table 1 highlight the functional relevance of emerging genetic and epigenetic markers in PCOS and provide a foundation for the subsequent sections of this review, which explore the broader implications of these molecular insights for disease pathophysiology and therapeutic innovation (Table 2).

### 3.2. Epigenetic Regulation and Environmental Interactions

Epigenetic mechanisms—such as DNA methylation, histone modifications, and non-coding RNAs—regulate gene expression without altering the underlying DNA sequence (Figure 2). These processes are dynamic, reversible, and responsive to external stimuli, including environmental exposures, dietary patterns, physical activity, stress, endocrine disruptors, and the hormonal milieu. In the context of PCOS, growing evidence implicates both global and gene-specific hypomethylation and hypermethylation events, particularly affecting key genes involved in steroidogenesis (e.g., *CYP19A1*), inflammation (e.g., *TNF-α*), and insulin signaling (e.g., *INSR*) [12,22,23]. These epigenetic changes can influence androgen production, ovarian folliculogenesis, metabolic homeostasis, and immune responses—hallmarks of PCOS pathophysiology.

Notably, differentially methylated regions (DMRs) have been identified in peripheral blood mononuclear cells (PBMCs), adipose tissue, ovarian theca cells, and granulosa cells of women with PCOS. Some of these DMRs correlate strongly with specific clinical phenotypes, including hyperandrogenism, anovulation, insulin resistance, and obesity, suggesting a potential role for epigenetic profiling as a diagnostic or prognostic tool. In particular, the altered expression of epigenetic regulators such as *DNMTs*, *HDACs*, and specific microRNAs (e.g., *miR-93*, *miR-223*) further supports the involvement of complex regulatory networks in PCOS.

Furthermore, epigenetic dysregulation may begin in utero. Maternal hyperandrogenemia, insulin resistance, and obesity during pregnancy have been shown to induce persistent epigenetic modifications in the fetus [24], which may predispose female offspring to PCOS-like features later in life. These findings lend strong support to the “Developmental Origins of Health and Disease” (DOHaD) hypothesis [25], which posits that adverse intrauterine exposures can have long-lasting effects on gene expression and disease susceptibility.

This transgenerational model suggests that epigenetic programming may serve as a biological bridge between maternal environment and offspring health, mediated by changes in chromatin structure, DNA methylation, and non-coding RNA expression. Importantly, Wadhwa et al. [25] highlighted how critical periods such as gestation, infancy, and puberty are particularly sensitive to environmental inputs, with long-term consequences for endocrine, metabolic, and reproductive health.

In light of these insights, the epigenome represents a promising therapeutic and preventive target in PCOS. Emerging strategies, including dietary interventions, physical activity, anti-inflammatory agents, and micronutrient supplementation (e.g., folate, vitamin D), may help modulate epigenetic patterns in a favorable direction. Moreover, the identification of stable, non-invasive epigenetic biomarkers in blood or follicular fluid could revolutionize early diagnosis, risk stratification, and individualized management in PCOS (Table 3).

### 3.3. Epigenetic Modifications in PCOS

The interplay between genetic variants and epigenetic modifications contributes to the heterogeneity observed in PCOS clinical presentations.

For instance, individuals carrying risk alleles in genes like *DENND1A* or *INSR* may exhibit varying degrees of symptom severity depending on their epigenetic landscape, shaped by environmental exposures and lifestyle factors [12,20,21].

This complex interaction underscores the need for personalized approaches in the diagnosis and management of PCOS.

Beyond genetic predisposition, epigenetic alterations have emerged as significant contributors to PCOS pathophysiology.

Some studies have demonstrated aberrant DNA methylation patterns and histone modifications in ovarian tissues of PCOS patients [26,27].

DNA Methylation: The hypomethylation of specific gene promoters, such as those regulating steroidogenesis and insulin signaling, has been observed, leading to dysregulated gene expression [27].

Histone Modifications: Alterations in histone acetylation and methylation states have been implicated in the transcriptional repression or activation of genes involved in ovarian function and metabolic processes [27,28].

These epigenetic changes may result from environmental factors, including diet, exposure to endocrine-disrupting chemicals, and lifestyle, further influencing the PCOS phenotype [28,29].

Recent investigations have highlighted the gut microbiota as a key contributor to the pathophysiology of polycystic ovary syndrome (PCOS) [29,30].

Beyond its established role in metabolic regulation and immune modulation, the gut microbiota also appears to exert significant influence on epigenetic mechanisms. Microbial metabolites—particularly short-chain fatty acids such as butyrate, acetate, and propionate—can modulate histone acetylation and DNA methylation patterns in host cells, including those involved in ovarian and endocrine function. These epigenetic alterations may influence gene expression related to insulin sensitivity, inflammation, and steroidogenesis, all of which are core features of PCOS. Dysbiosis, or imbalance in gut microbial composition, may thus not only trigger systemic metabolic disturbances but also promote heritable epigenetic changes that contribute to disease onset and persistence. This emerging evidence supports the concept of a microbiota–epigenome–ovary axis in PCOS, emphasizing the potential of microbiome-targeted interventions to modify disease trajectory through both direct metabolic effects and indirect epigenetic reprogramming [30].

Women with PCOS exhibit reduced microbial diversity and a lower abundance of beneficial bacterial species compared to healthy controls.

Several mechanisms have been proposed to explain this association, including increased intestinal permeability and systemic endotoxemia, the altered production of short-chain fatty acids (SCFAs), leading to impaired insulin sensitivity and the disruption of bile acid metabolism and hormonal regulation [30,31].

Strikingly, fecal microbiota transfer from PCOS-affected individuals into germ-free mice has been shown to induce both metabolic and reproductive abnormalities consistent with PCOS phenotypes, supporting a causal role for gut dysbiosis. In light of this, microbiota-targeted interventions—including probiotics, prebiotics, and fecal microbiota transplantation (FMT)—are currently under investigation as potential therapeutic strategies [32].

Alterations in gut microbial composition have been consistently observed in women with PCOS, characterized by a reduction in beneficial bacterial taxa such as *Lactobacillus* and *Bifidobacterium*, alongside an increase in pro-inflammatory genera like *Escherichia* and *Shigella*. These dysbiotic shifts are closely associated with hallmark features of PCOS, including insulin resistance and hyperandrogenism. The emerging concept of a “gut–ovary axis” suggests a bidirectional interaction whereby gut microbiota may actively contribute to the pathogenesis of PCOS, rather than merely reflecting its metabolic disturbances. This growing evidence supports the exploration of microbiota-targeted interventions—such as probiotics and dietary modifications—as promising adjuncts in the therapeutic approach to PCOS [30,31,32].

### 3.4. Future Directions

Advances in the understanding of the genetic, epigenetic, and microbial underpinnings of polycystic ovary syndrome (PCOS) have opened new avenues for the development of precision medicine strategies tailored to individual patients [33].

Several promising directions are currently being explored:−Pharmacogenomics: Genetic profiling of patients—particularly those harboring variants in genes involved in insulin signaling or androgen biosynthesis—may inform the selection of therapeutic agents. For instance, the identification of insulin resistance (IR)-associated polymorphisms may help predict responsiveness to insulin sensitizers such as metformin or inositol-based therapies [34].−Epigenetic therapy: Although still in preclinical stages, targeting epigenetic regulators offers an exciting therapeutic frontier. Inhibitors of histone deacetylases (HDACs) and DNA methyltransferases (DNMTs) have shown potential in reversing abnormal gene expression patterns associated with PCOS in experimental models, suggesting their future utility in modulating disease-relevant pathways [35,36].−Microbiota modulation: Given the emerging role of gut dysbiosis in PCOS pathophysiology, interventions aimed at restoring eubiosis hold considerable promise. Dietary modification, prebiotic and probiotic supplementation, and fecal microbiota transplantation (FMT) are being investigated for their capacity to attenuate inflammation, improve metabolic function, and possibly restore hormonal balance [29,30].

Looking forward, the integration of multi-omics data—including genomics, epigenomics, transcriptomics, and metagenomics—may enable the development of fully personalized treatment paradigms. Such approaches have the potential not only to improve therapeutic efficacy and minimize adverse effects but also to identify at-risk individuals before the onset of clinical symptoms.

## 4. Discussion

Polycystic ovary syndrome (PCOS) is increasingly recognized as a multifactorial endocrine and metabolic disorder with a complex polygenic architecture, shaped by intricate interactions between genetic, epigenetic, and environmental determinants. Its highly heterogeneous clinical presentation—including hyperandrogenism, oligo-anovulation, and polycystic ovarian morphology—cannot be fully explained by inherited susceptibility alone. Rather, it results from dynamic and reciprocal interactions with socioeconomic conditions, geographic and lifestyle factors, and environmental exposures, particularly in regions characterized by elevated levels of industrial pollutants and endocrine-disrupting chemicals [37].

In the context of our review, well-established genetic contributors to PCOS—such as defects in androgen biosynthesis (e.g., *CYP11A1*, *CYP17A1*), abnormalities in insulin signaling (e.g., *INSR*, *IRS1*), and alterations in gonadotropin receptor function (e.g., *FSHR*, *LHCGR*)—were not discussed in detail [12,20,21]. This was a deliberate methodological choice, made in light of the vast amount of existing literature on these canonical genes. Instead, we prioritized the investigation of more recent and less-characterized mechanisms, particularly those related to gene expression regulation, epigenetic modifications, and the emerging role of non-coding RNAs. This approach aimed to capture novel molecular insights with potential translational relevance. Nonetheless, to acknowledge the foundational importance of these genes in PCOS pathogenesis, we have included a dedicated summary table (Table 3) within the manuscript, which presents their biological functions and associated molecular pathways in a structured and comparative format [2,7,24].

Seminal twin and family studies, such as that conducted by Vink et al. [20], have highlighted the substantial heritability of PCOS, supporting the involvement of both autosomal and X-linked inheritance patterns. Genetic contributors include variants in genes regulating androgen biosynthesis and action (e.g., *CYP11A1*, *CYP17A1*, *DENND1A*), insulin signaling (e.g., INSR, IRS1), and gonadotropin function (e.g., *FSHR*, *LHCGR*) [12,20,21].

Genome-wide association studies (GWAS) have expanded this list, identifying novel loci such as *THADA*, *TOX3*, *YAP1*, and *AMH/AMHR2* that influence diverse aspects of PCOS pathophysiology, including folliculogenesis, metabolic regulation, and hormonal homeostasis [12,37,38,39].

While these variants often exhibit modest individual effect sizes, their cumulative impact substantially increases disease risk. Importantly, gene–environment interactions may modulate both the penetrance and expressivity of these variants, emphasizing the need for population-specific genomic studies, especially in underrepresented ethnic groups where allele frequencies and phenotypic manifestations may diverge [40].

The advent of next-generation sequencing (NGS) technologies has markedly accelerated the discovery of rare and common variants associated with PCOS, enabling more comprehensive, cost-effective screening compared to traditional Sanger sequencing [39].

Dapas and Dunaif [41] critically reviewed the genetic architecture of polycystic ovary syndrome (PCOS), integrating genome-wide association study (GWAS) findings with clinical phenotypes. They proposed a stratified model of PCOS based on distinct genetic and molecular signatures, suggesting that diverse causal mechanisms underlie its heterogeneous manifestations. This genomic framework offers novel insights into PCOS pathophysiology and may guide future subclassification and personalized therapeutic strategies [41].

These advances have not only facilitated the identification of novel candidate genes but also improved our capacity to functionally characterize molecular pathways implicated in the disorder [42]. Beyond inherited genetic variation, accumulating evidence underscores the pivotal role of epigenetic mechanisms in modulating gene expression in response to internal and external stimuli [43].

Aberrant DNA methylation patterns, histone modifications, and the altered expression of non-coding RNAs contribute to the disruption of metabolic and reproductive pathways in PCOS. The hypomethylation of key regulatory genes has been associated with chronic low-grade inflammation, dysregulated steroidogenesis, and abnormal insulin signaling, distinguishing PCOS patients from healthy controls and potentially mediating the impact of environmental insults on disease development [44].

Notably, many of these epigenetic alterations have been detected not only in ovarian tissue but also in accessible peripheral compartments such as blood and saliva, supporting their potential utility as non-invasive biomarkers for early detection, disease stratification, and monitoring of therapeutic response [45].

Recent investigations [11,14] have brought increasing attention to the role of non-coding RNAs—including long non-coding RNAs (lncRNAs) such as *SNHG5*, circular RNAs, and microRNAs—in regulating granulosa cell proliferation, oocyte competence, and endometrial receptivity. These post-transcriptional regulatory networks add further complexity to the molecular landscape of PCOS and may hold the key to understanding differential responses to treatment, especially in the context of assisted reproductive technologies [46].

The epigenetic regulation of inflammatory mediators and insulin signaling genes, such as *TGF-β1*, *FoxO1*, and components of the *TLR4/NF-κB/NLRP3* inflammasome axis, further highlights the interdependence of genetic susceptibility and environmental exposure in shaping disease progression and phenotypic variability [14,15,16].

Future directions in PCOS research and clinical translation will benefit from the integration of multi-omics technologies—combining genomics, transcriptomics, epigenomics, proteomics, and metabolomics—to generate a systems-level understanding of disease pathogenesis [47]. Such an approach may enable the delineation of distinct molecular subtypes of PCOS and support the development of personalized diagnostic and therapeutic strategies. Longitudinal cohort studies are urgently needed to elucidate the temporal relationships between early molecular perturbations and clinical features. In parallel, functional studies employing *CRISPR-Cas9* gene editing, transcriptomic profiling, and iPSC-derived ovarian and endometrial models offer powerful platforms for validating candidate gene functions and dissecting complex gene regulatory networks [48].

A particularly urgent area of investigation is the development of reliable, minimally invasive biomarkers for early diagnosis, prognosis, and therapeutic monitoring [49].

Epigenetic signatures detectable in circulating biofluids hold great promise in this regard, potentially enabling earlier intervention and the more precise management of PCOS subtypes. Advances in molecular understanding are also poised to inform the design of targeted therapies aimed at modulating insulin resistance, inflammatory responses, or epigenetic dysregulation—thereby moving beyond symptomatic treatment to potentially modifying the disease trajectory itself [50].

Emerging data on developmental origins of health and disease suggest that prenatal and early-life exposures—including maternal hyperandrogenism, gestational diabetes, and exposure to endocrine disruptors—can induce lasting epigenetic reprogramming that predisposes offspring to PCOS. Such findings not only provide mechanistic insights into the transgenerational transmission of disease risk but also highlight critical windows for early preventive intervention [25].

Understanding how these early-life events interact with genetic predisposition to shape adult phenotypes remains a pressing challenge and a promising avenue for future research [51,52].

In conclusion, PCOS is a highly prevalent and heterogeneous disorder arising from the convergence of multiple genetic, epigenetic, and environmental influences. Although significant progress has been made in identifying susceptibility loci and molecular pathways involved in its pathogenesis, many aspects remain poorly understood. Addressing these knowledge gaps will require sustained, interdisciplinary collaboration across reproductive endocrinology, molecular genetics, environmental health, and computational biology. Only through such integrative efforts can we hope to advance precision medicine approaches that improve outcomes for individuals affected by this complex and burdensome condition.

### Strengths and Limitations

This study offers a comprehensive synthesis of current genetic and epigenetic findings related to the pathogenesis of PCOS, integrating data from candidate gene studies, GWAS, and epigenomic profiling. One of its main strengths lies in the multidimensional approach, which considers both hereditary and environmental influences, highlighting the complexity of gene–environment interactions. Furthermore, the discussion of emerging non-coding RNA mechanisms and potential biomarkers provides a forward-looking perspective with translational relevance.

However, several limitations must be acknowledged. First, while the review draws from a broad range of sources, it is not a systematic review and may be subject to selection bias. Second, many of the cited studies were conducted in specific populations, limiting generalizability across diverse ethnic groups. Third, although genetic associations are discussed extensively, the functional validation of many candidate genes remains incomplete, and causality cannot yet be firmly established.

Finally, despite the growing interest in epigenetic modifications, the field remains in its early stages, and further research is needed to clarify the temporal dynamics and tissue specificity of these changes in PCOS.

Moreover, this review is intended to lay the groundwork for future research, fostering investigations with larger and more diverse populations to validate and expand upon these preliminary molecular insights.

As a final consideration, it is worth reflecting on the importance of integrating rigorous study selection with scientific innovation.

The limited number of studies ultimately included in this review stems from a deliberate and methodologically robust selection strategy, aimed at identifying works that offer truly novel and mechanistically informative contributions to the genetic and epigenetic understanding of PCOS. While this approach inevitably excluded many broadly associative or repetitive studies, it also represents a strength: by prioritizing originality and translational relevance, this review delivers a focused and impactful synthesis of the most meaningful recent advances in the field.

## 5. Conclusions

Polycystic ovary syndrome (PCOS) is a multifactorial endocrine and metabolic disorder shaped by intricate genetic, epigenetic, and environmental interactions.

Despite substantial advances in elucidating susceptibility loci and underlying molecular mechanisms, the etiopathogenesis of PCOS remains only partially understood. Addressing its clinical heterogeneity and improving patient outcomes will require sustained, multidisciplinary collaboration across reproductive endocrinology, molecular genetics, epigenomics, bioinformatics, and environmental health sciences.

Future research efforts should prioritize integrative, systems-level approaches, alongside functional and longitudinal studies, to enable earlier diagnosis, more precise phenotypic classification, and the development of targeted, disease-modifying therapies. Such integrative strategies hold the promise of transforming PCOS management from symptomatic relief to true precision medicine.

## Figures and Tables

**Figure 1 biomedicines-13-01912-f001:**
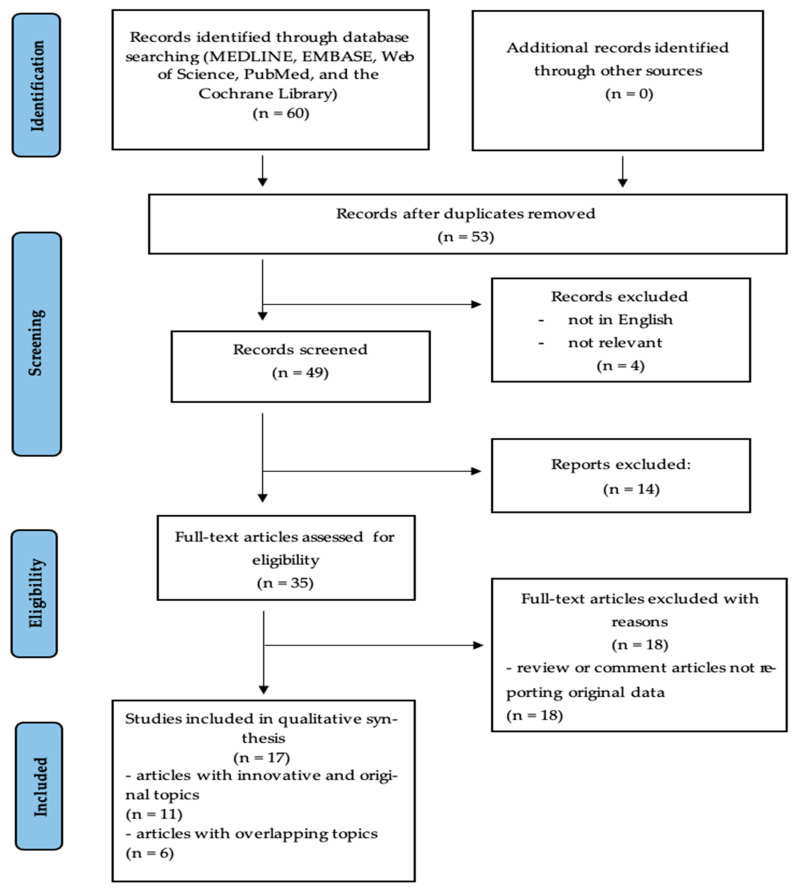
Flow diagram of this review search.

**Figure 2 biomedicines-13-01912-f002:**
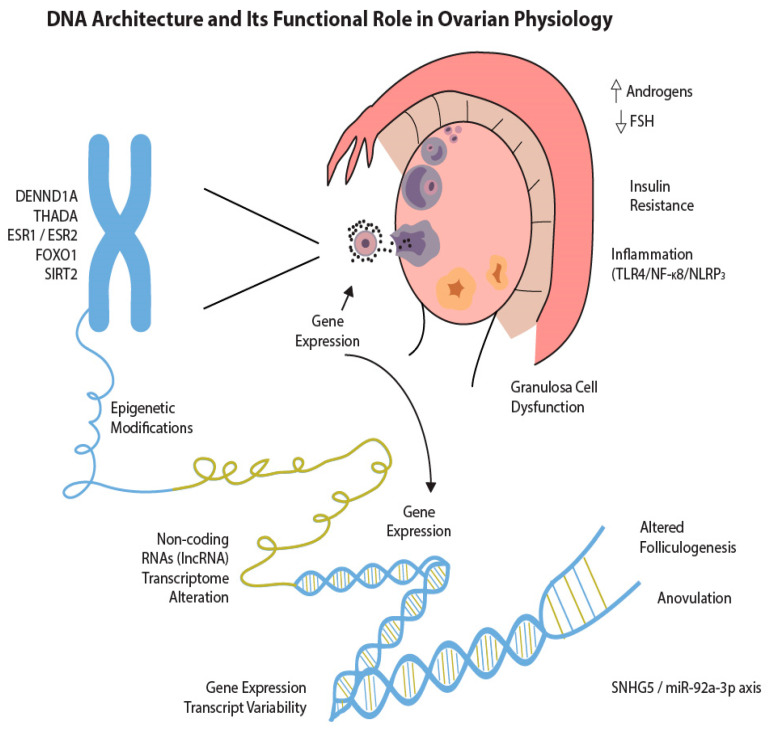
DNA architecture and its functional role in ovarian physiology. ↑ indicates increase; ↓ indicates decrease.

**Table 1 biomedicines-13-01912-t001:** Representative studies on PCOS-associated genes: insights into molecular mechanisms and clinical relevance.

Author (Year)	Population	Sample Size	Age (Years)	BMI/Weight	Gene(s)	Method	Type of Analysis	Main Outcome	Key Findings
González-Fernández et al., 2019 [9]	Human	16 (case), 24 (control)	27–39	NR	*SIRT1–7*	RT-PCR	Gene expression	Association between *SIRT7* and PCOS	*SIRT2* overexpression
Mucee et al., 2024 [10]	Human	15 (*ESR1*), 15 (*ESR2*)	NR	NR	*ESR1, ESR2*	Bioinformatics	SNP analysis	Correlation of SNPs with PCOS	10 SNPs showed strong associations
Akbari et al., 2023 [11]	Human	33 (case), 33 (control)	31.1 ± 5.2 vs. 33.8 ± 5.3	66.4 ± 7.7 vs. 66.0 ± 6.4	*CALM1, PSMD6, AK124742*	RT-PCR	Gene expression	Expression in cumulus cells (CCs) of PCOS patients	↑ *CALM1*, ↓ *PSMD6* and *AK124742*
Yu et al., 2023 [12]	Human	2504	NR	NR	37 SNPs (incl. *DENND1A, AOPEP, THADA, DGKI, UNC5C*)	RT-PCR	SNP analysis	Genetic selection in PCOS	Positive selection for 5 genes
Zhao et al., 2024 [13]	Mouse	12 (case), 12 (control)	NA	NA	205 circRNAs	Microarray	Gene expression	Endometrial and ovarian dysfunction	147 upregulated, 58 downregulated circRNAs
Jiang et al., 2024 [14]	Human	15 (case), 15 (control)	29.8 (25–33) vs. 28.3 (20–36)	23.5 ± 3.4 vs. 26.3 ± 5.0	*ANXA3, CXCR2, IQGAP2, LMNB1*	RT-PCR	Gene expression	Aging-related DEGs in PCOS	73 aging-related DEGs identified
Huang et al., 2024 [15]	Mouse	20 (case), 20 (control)	NA	NA	*FOXO1*	Western blot, ELISA, cytometry	Gene expression	*FOXO1* and PCOS correlation	*FOXO1* upregulation in PCOS
Gao et al., 2024 [16]	Human	PCOS “Han” families	NR	NR	*TGF-β1*	qPCR	DNA methylation	CpG methylation and PCOS phenotype	Hypomethylation associated with PCOS
Monshizadeh et al., 2024 [17]	Human	38 (case), 33 (control)	32.58 ± 5.48 vs. 34.50 ± 4.00	28.16 ± 5.53 vs. 26.55 ± 3.94	*RBX1, BAMBI*	RT-PCR	Gene expression	Expression in MII and GV cumulus cells	↓ *RBX1* and *BAMBI* in PCOS patients
Postolache et al., 2024 [18]	Human	212 families	NR	NR	*MTNR1A, MTNR1B*	Genotyping	SNP analysis	*MTNR* variants and PCOS risk	4 variants (*MTNR1A*), 2 variants (*MTNR1B*) linked to PCOS
Yang et al., 2024 [19]	Human, mouse, in vitro	Cohort 1 30 (case), 30 (control) Cohort 2 60 (case), 60 (control)	NR	NR	*SNHG5*	RT-PCR	Gene expression	*SNHG5* and follicular development	*SNHG5* suppresses follicular growth via *miR-92a-3p/CDKN1C* axis

↑ indicates increase; ↓ indicates decrease; NA, not available; NR, not reported.

**Table 2 biomedicines-13-01912-t002:** Summary of key PCOS-related genes and their functions. The table outlines 20 genes linked to PCOS, highlighting their types, biological functions, and involvement in relevant pathways or GO terms.

Gene	Gene Type	Biological Function	Associated Pathways/Gene Ontology Terms Annotations
*SIRT* [9]	Protein coding	Exhibits mono-ADP ribosyltransferase or deacylase activity; involved in oxidative stress, autophagy, ovulation disturbances, and insulin resistance.	Cellular response to oxidative stress, metabolic regulation.
*ESR1/2* [10]	Protein coding	Encode estrogen receptors that bind and mediate the effects of estrogen.	Estrogen signaling pathway.
*CALM1* [11]	Protein coding	Calmodulin regulates calcium signal transduction, ion channels, enzymes, and aquaporins.	Calcium signaling pathway.
*PSMD6* [11]	Protein coding	Part of the 26S proteasome complex responsible for ATP-dependent degradation of ubiquitinated proteins.	Protein degradation via ubiquitin-proteasome pathway.
*AK124742* [11]	Long non-coding RNA (lncRNA)	Regulates gene expression, associated with embryo quality and pregnancy outcomes.	Gene expression regulation (lncRNA-mediated).
*DENND1A* [12,20,21]	Protein coding	Guanine nucleotide exchange factor involved in vesicle-mediated transport.	Vesicle-mediated transport, Rab regulation of trafficking; GO: SH3 domain binding, GEF activity.
*AOPEP* [12]	Protein coding	Zinc-dependent aminopeptidase that removes N-terminal amino acids; involved in angiotensin IV generation.	Metallopeptidase activity, blood pressure regulation.
*THADA* [12]	Protein coding	Methylates the 2′-O-ribose of tRNA; involved in tRNA modification.	RNA methylation.
*DGKI* [12]	Protein coding	Diacylglycerol kinase involved in converting diacylglycerol to phosphatidic acid.	Lipid signaling pathways.
*UNC5-family* [12]	Protein coding	Netrin receptor involved in axon guidance and cell migration during neural development.	Netrin signaling, cell migration.
*ANXA3* [14]	Protein coding	Calcium-dependent phospholipid-binding protein involved in inflammation and cancer.	Prostaglandin synthesis and regulation; GO: calcium ion binding.
*CXCR2* [14]	Protein coding	Chemokine receptor involved in immune cell migration and inflammation.	GPCR signaling, chemokine-mediated signaling; GO: C-X-C chemokine receptor activity.
*IQGAP* [14]	Protein coding	Scaffold protein involved in cytoskeletal regulation, cell adhesion, signaling, and antiviral responses.	Cytoskeletal regulation, antiviral innate immunity.
*LMNB1* [14]	Protein coding	Structural component of the nuclear lamina, involved in chromatin organization and apoptosis.	Apoptosis signaling, structural molecule activity; GO: phospholipase binding.
*FOXO1* [15]	Protein coding	Transcription factor involved in metabolism, apoptosis, and cell cycle regulation.	FOXO-mediated transcription, IL-9 signaling; GO: DNA-binding transcription factor activity.
*TGF-β1* [16]	Protein coding	Cytokine that controls cell growth, proliferation, differentiation, and apoptosis.	TGF-beta signaling pathway.
*RBX1* [17]	Protein coding	Involved in ubiquitination and cell cycle progression; contains a RING finger domain.	Ubiquitin-proteasome pathway.
*BAMBI* [17]	Protein coding	Pseudoreceptor for TGF-β; modulates TGF-beta signaling.	TGF-beta signaling modulation.
*MTNR1B* [18]	Protein coding	G-protein-coupled receptor for melatonin, involved in circadian rhythm regulation.	Melatonin signaling, circadian regulation.
*SNHG5* [19]	Long non-coding RNA (lncRNA)	Acts as a sponge for microRNAs and stabilizes mRNAs; involved in gene expression regulation.	lncRNA-mediated gene regulation, microRNA interaction.

**Table 3 biomedicines-13-01912-t003:** Key genes involved in PCOS etiopathogenesis: a functional categorization.

Functional Category	Gene	Biological Role	Mechanistic Relevance in PCOS
Androgen Biosynthesis	*CYP11A1* [12,20,21]	Initiates steroidogenesis by converting cholesterol to pregnenolone	Overexpression leads to androgen excess and theca cell hyperactivity
	*CYP17A1* [12,20,21]	Catalyzes 17α-hydroxylase and 17,20-lyase reactions in steroid biosynthesis	SNPs linked to hyperandrogenism and anovulation
Insulin Signaling and Resistance	*INSR* [12,20,21,22,23]	Insulin receptor activating PI3K-Akt signaling	Mutations cause insulin resistance and contribute to metabolic PCOS phenotype
	*IRS1* [12,20,21]	Adapter protein transmitting insulin/IGF-1 signals	Variants linked to impaired glucose uptake and metabolic syndrome in PCOS
Gonadotropin Response	*FSHR* [12,20,21]	FSH receptor regulating follicular maturation	Polymorphisms affect ovarian sensitivity and folliculogenesis
	*LHCGR* [12,20,21]	LH receptor essential for ovulation and corpus luteum maintenance	Mutations impair ovulatory response and promote androgen production

## Data Availability

The present review was based on published articles. All summary data generated during this study are included in this published article. Raw data used for the analyses are available presented in the original reviewed articles.

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
