# Peer review of "PCOS and the Genome: Is the Genetic Puzzle Still Worth Solving?"

_biomedicines, 2025, doi:10.3390/biomedicines13081912_

Round 1
Reviewer 1 Report
Comments and Suggestions for Authors
This review offers a comprehensive and well-organized synthesis of current genetic and epigenetic insights into polycystic ovary syndrome. The manuscript integrate findings from genomic, transcriptomic, and epigenomic studies, and contextualizing these within the broader pathophysiological framework of PCOS. While several issues warrant revision.
1. Authors were strongly encouraged to register their detailed protocols before data extraction commences, in a public registry such as PROSPERO (https://www.crd.york.ac.uk/prospero/). Authors must include registration information in the Methods section.
2. Some genomic research on PCOS are missing, such as PMID: 36720993 and PMID: 38632610
3. Refs. 4 and 35 are duplicated.
Refs. 5 and 30 are duplicated.
Refs. 34 and 40 are duplicated.
4. Some recent studies related to Microbiota–Gut–Ovary Axis may further support part 3.4 (e.g., PMID: 37559119).
5. Line 31: "pa erns" → should be corrected to "patterns".
Author Response
July 23, 2025
Dear Editors,
We received the reviewer’s comments regarding our manuscript submitted for consideration for publication entitled "PCOS and the Genome: Is the Genetic Puzzle Still Worth Solving?” and we are grateful for the opportunity to revise our work. We would like to thank the reviewers for considering the work interesting and for taking the time to make those very appropriate comments to improve it. We have modified the text of the manuscript to address all the areas identified by the reviewers. We believe these revisions have adequately addressed the raised points, and we hope that the revised version of the manuscript is now considered acceptable for publication.
ACCORDING TO REVIEWER 1’S SUGGESTIONS:
This review offers a comprehensive and well-organized synthesis of current genetic and epigenetic insights into polycystic ovary syndrome. The manuscript integrate findings from genomic, transcriptomic, and epigenomic studies, and contextualizing these within the broader pathophysiological framework of PCOS. While several issues warrant revision.
- Authors were strongly encouraged to register their detailed protocols before data extraction commences, in a public registry such as PROSPERO (https://www.crd.york.ac.uk/prospero/). Authors must include registration information in the Methods section.
1) We thank the Reviewer for the valuable suggestion. In accordance with the journal’s guidelines, the review has now been prospectively registered in the PROSPERO database prior to data extraction. The registration ID is CRD420241110847, and the relevant information has been added to the Methods section of the revised manuscript.
- Some genomic research on PCOS are missing, such as PMID: 36720993 and PMID: 38632610
2) We thank the Reviewer for highlighting these relevant references. In response, we have included both suggested studies (PMID: 36720993 and PMID: 38632610) in the revised manuscript. These papers have been integrated into the text and properly cited in the reference list, contributing to a more comprehensive overview of the genomic landscape in PCOS.
- Refs. 4 and 35 are duplicated.
Refs. 5 and 30 are duplicated.
Refs. 34 and 40 are duplicated.
3) Thank you for pointing that out. I have reviewed the reference list and have removed the duplicate entries corresponding to references 4 and 35, 5 and 30, and 34 and 40. The updated reference list now includes only unique citations.
- Some recent studies related to Microbiota–Gut–Ovary Axis may further support part 3.4 (e.g., PMID: 37559119).
4) Thank you for the suggestion regarding the inclusion of recent studies on the Microbiota–Gut–Ovary Axis (e.g., PMID: 37559119). I have integrated this reference into the manuscript, incorporating it within the preceding paragraph as recommended by the other reviewer to enhance flow and coherence.
- Line 31: "pa erns" → should be corrected to "patterns".
5) The typo “pa erns” has been corrected to “patterns.”
Reviewer 2 Report
Comments and Suggestions for Authors
The present paper is an interesting and well-written one focused on the genetics of PCOS, and especially on some newly discovered genetic factors such as dysregulation of sirtuins and the role of inflammation-related signaling pathways. The paper would be of interest for the readers so I think it might be suitable for publication after addressing some important concerns:
1.The authors aimed “to provide an updated overview of the current evidence regarding the role of genetic variants the etiopathogenesis of PCOS, with a focus on their impact on ovarian function, fertility, and systemic alterations”. However, the paper found only 60 papers associated with PCOS genetics, and excluded 43, which is very strange considering the huge number of papers studying genetic SNP polymorphisms in PCOS. There is no time period for search /defined in the method section/, though most papers cited have been published in the last 5 years. There are similar reviews investigating the role of different genetic factors and the authors emphasize on the “genetic contributors regulating androgen biosynthesis and action (e.g., CYP11A1, CYP17A1, insulin signaling (e.g., INSR, IRS1), and gonadotropin function (e.g., FSHR, LHCGR), but these genetic factors are not included in the paper. Thus, a table with all these genetic factors associated with PCOS should be added, and the reasons for selecting specific genes, e.g., functional analysis/gene expression, should be clearly stated in the text.
2.The part focused on the Microbiota–Gut–Ovary Axis is irrelevant to the aims of the paper.
3.The included references should be cited more carefully, e.g., it is stated that the number of patients is not reported by González-Fernández et al., 2019. Actually, it has been reported - 16 PCOS patients compared to 24 women without ovarian dysfunction but tubal or male factor The same for Yang et al., 2024 (30/30). Thus, all references should be checked and cited properly. Double references should be removed.
4.The very low number of the investigated PCOS patients by the selected studies (González-Fernández et al., 2019; Mucee et al., 2024) limits the conclusions and should be included as an important limitation.
Author Response
July 23, 2025
Dear Editors,
We received the reviewer’s comments regarding our manuscript submitted for consideration for publication entitled "PCOS and the Genome: Is the Genetic Puzzle Still Worth Solving?” and we are grateful for the opportunity to revise our work. We would like to thank the reviewers for considering the work interesting and for taking the time to make those very appropriate comments to improve it. We have modified the text of the manuscript to address all the areas identified by the reviewers. We believe these revisions have adequately addressed the raised points, and we hope that the revised version of the manuscript is now considered acceptable for publication.
ACCORDING TO REVIEWER 2’S SUGGESTIONS:
The present paper is an interesting and well-written one focused on the genetics of PCOS, and especially on some newly discovered genetic factors such as dysregulation of sirtuins and the role of inflammation-related signaling pathways. The paper would be of interest for the readers so I think it might be suitable for publication after addressing some important concerns:
1.The authors aimed “to provide an updated overview of the current evidence regarding the role of genetic variants the etiopathogenesis of PCOS, with a focus on their impact on ovarian function, fertility, and systemic alterations”. However, the paper found only 60 papers associated with PCOS genetics, and excluded 43, which is very strange considering the huge number of papers studying genetic SNP polymorphisms in PCOS. There is no time period for search /defined in the method section/, though most papers cited have been published in the last 5 years. There are similar reviews investigating the role of different genetic factors and the authors emphasize on the “genetic contributors regulating androgen biosynthesis and action (e.g., CYP11A1, CYP17A1, insulin signaling (e.g., INSR, IRS1), and gonadotropin function (e.g., FSHR, LHCGR), but these genetic factors are not included in the paper. Thus, a table with all these genetic factors associated with PCOS should be added, and the reasons for selecting specific genes, e.g., functional analysis/gene expression, should be clearly stated in the text.
1) We fully acknowledge the reviewer’s concern regarding the number of included studies.
As specified in the revised Materials and Methods section, our literature search covered the period from January 2015 to June 2025, and focused specifically on studies reporting functional, mechanistic, or transcriptomic evidence, rather than solely associative SNP studies.
This methodological choice was deliberate and aligned with the objective of offering an innovative and mechanistically informative synthesis. Many broadly associative or repetitive studies were therefore excluded to enhance the translational relevance and originality of the review, as discussed in the Strengths and Limitations section. That said, we appreciate the reviewer’s suggestion and have addressed it by adding a new Table 3, which includes the molecular mechanisms of the most frequently reported PCOS-related genes (including CYP11A1, CYP17A1, INSR, IRS1, FSHR, LHCGR), along with their biological functions and associated pathophysiological pathways. These genes are now also mentioned and discussed in the updated Results and Discussion sections, particularly in relation to their functional significance in steroidogenesis, insulin signaling, and gonadotropin activity.
Moreover, the rationale for our gene selection strategy—favoring studies with gene expression data, epigenetic profiling, or functional validation—has been more clearly articulated in both the Materials and Methods and Results sections.
Once again, we thank the reviewer for these valuable suggestions, which have helped improve the scope, clarity, and scientific depth of the manuscript.
2.The part focused on the Microbiota–Gut–Ovary Axis is irrelevant to the aims of the paper.
2) We thank the reviewer for this observation. In response to the suggestion, we have removed the dedicated subparagraph on the Microbiota–Gut–Ovary Axis, as recommended. However, to maintain coherence with the broader etiopathogenetic framework and to address specific references suggested by another reviewer (e.g., PMID: 37559119), we have integrated the most relevant concepts into the preceding section on epigenetic mechanisms. This integration allowed us to contextualize emerging evidence on how the gut microbiota may influence PCOS through epigenetic regulation, in line with the manuscript’s focus on molecular and functional mechanisms.
We believe this adjustment preserves the scientific value of the referenced studies while aligning more closely with the paper’s primary objectives.
3.The included references should be cited more carefully, e.g., it is stated that the number of patients is not reported by González-Fernández et al., 2019. Actually, it has been reported - 16 PCOS patients compared to 24 women without ovarian dysfunction but tubal or male factor The same for Yang et al., 2024 (30/30). Thus, all references should be checked and cited properly. Double references should be removed.
3) We thank the reviewer for the valuable observation. In response, we have not only corrected the specific references mentioned (González-Fernández et al., 2019 and Yang et al., 2024), but we have also performed a comprehensive check of all included references to ensure the accuracy of patient numbers, methodologies, and outcomes reported in the text/table.
In addition, duplicate references have been identified and removed, and all citations have been revised to align with the journal’s citation standards.
We are confident that the manuscript now reflects a more precise and rigorous use of the referenced literature, and we appreciate the reviewer’s contribution to strengthening the quality of the paper.
4.The very low number of the investigated PCOS patients by the selected studies (González-Fernández et al., 2019; Mucee et al., 2024) limits the conclusions and should be included as an important limitation.
4) We thank the reviewer for this important observation. We have added the suggested point in the “Strengths and Limitations” section, explicitly acknowledging that the small sample sizes in several of the included studies (e.g., González-Fernández et al. [9], Mucee et al. [10]) may limit the statistical power and generalizability of their findings.
However, we also wish to clarify that the inclusion of these studies was a deliberate choice: we aimed to explore specific molecular mechanisms and genes that, although based on smaller cohorts, were among the least discussed and least represented in the existing literature. Our intention was to highlight emerging and underexplored molecular pathways in PCOS, offering potentially novel insights. We sincerely hope that our effort to bring attention to these overlooked aspects may be appreciated by the reviewer.
Once again, we would like to thank the Reviewers for the precious suggestions and the Editor for allowing us to improve our manuscript. We hope our work now has the quality to be accepted for publication in your prestigious Journal.
We remain at your disposal for any further detail you might consider essential to clarify.
On behalf of the co-Authors,
Mario Palumbo
MD, phD cand.
University of Naples “Federico II”
mpalumbomed@gmail.com
mario.palumbo@unina.it
Round 2
Reviewer 2 Report
Comments and Suggestions for Authors
The paper is improved and some methodological limitations have been included,
however, several minor comments should addressed before publication:
- time limit of search /2015-2025/ should be included in the abstract;
- tables 2 and 3 need proper references citations.
Author Response
August 2, 2025
Dear Editors,
We have received the reviewer’s comments regarding our manuscript entitled "PCOS and the Genome: Is the Genetic Puzzle Still Worth Solving?”, submitted for consideration for publication. We are grateful for the opportunity to revise our work and would like to thank the reviewers for finding our manuscript of interest and for their valuable suggestions to improve it.
We have modified the manuscript to address all the points raised. We believe that the revised version adequately responds to the comments and we hope it is now suitable for publication.
ACCORDING TO REVIEWER 2’S SUGGESTIONS (Round 2):
“The paper is improved and some methodological limitations have been included. However, several minor comments should be addressed before publication:
- Time limit of search (2015–2025) should be included in the abstract.
- Tables 2 and 3 need proper reference citations.”
Response to Reviewer:
We sincerely thank the reviewer for the constructive feedback. We have addressed all comments as follows:
- Time limit of search (2015–2025) in the abstract:
The time frame of the literature search (2015–2025) has now been explicitly included in the abstract to enhance clarity and methodological transparency. - Reference citations for Tables 2 and 3:
Proper reference citations have been added in the footnotes of Tables 2 and 3, in accordance with the reviewer’s recommendation.
We are grateful for the opportunity to improve our manuscript and sincerely thank both the reviewers and the Editor for their thoughtful suggestions. We hope that the revised version meets the Journal’s standards and can now be considered for publication.
Please do not hesitate to contact us should any further clarification be needed.
On behalf of all co-authors,
Mario Palumbo
MD, PhD Candidate
Department of Public Health
University of Naples “Federico II”
mpalumbomed@gmail.com
mario.palumbo@unina.it
